# Does Systemic Methotrexate Therapy Induce Azole Resistance among Endogenous *Candida* Strains?

**DOI:** 10.3390/antibiotics10111302

**Published:** 2021-10-26

**Authors:** Dawid Żyrek, Joanna Nowicka, Magdalena Pajączkowska, Ewa Morgiel

**Affiliations:** 1Department of Microbiology, Faculty of Medicine, Wrocław Medical University, 50-367 Wrocław, Poland; magdalena.pajaczkowska@umed.wroc.pl; 2Department of Rheumatology and Internal Medicine, Faculty of Medicine, Wrocław Medical University, 50-367 Wrocław, Poland; ewa.morgiel@gmail.com

**Keywords:** *Candida*, cross-resistance, methotrexate, fluconazole

## Abstract

Background: Research confirms that *Candida* spp. incubated with methotrexate develop multi-drug resistance to azoles, but it is not clear whether this phenomenon occurs in vivo in patients treated with cytostatics. The aim of the study was to assess whether systemic methotrexate therapy induces resistance to azoles among endogenous *Candida* strains in patients with rheumatological diseases. Methods: The test group consisted of 52 rheumatological patients on methotrexate therapy, who have never been exposed to fluconazole. The control group was composed of 49 individuals who have never been exposed to either methotrexate or fluconazole. Oral swab and clinical information were obtained from each participant. The acquired material was cultured, then each strain was isolated and identified (MALDI TOF). Subsequently, minimal inhibitory concentration (MIC) for fluconazole was determined. Results: MIC values ranged from <0.125 to 64 µg/mL with the most common result <0.125 µg/mL. Samples obtained from 4 patients of the test group and 2 patients of the control group contained strains resistant to fluconazole. Conclusions: Despite slightly higher incidence of fluconazole-resistant strains among patients on systemic methotrexate therapy, we found no solid evidence to support the hypothesis that methotrexate induces resistance to azoles among endogenous *Candida* strains in patients with rheumatological diseases.

## 1. Introduction

The use of cytostatic drugs is widespread in many fields of medicine, among others, in oncology, haematology, and rheumatology. By reducing leukocyte cell division, their action disrupts the functioning of the immune system, which significantly increases the chances of symptomatic infections caused by endogenous microorganisms, including fungi [1,2]. One of the most common side effects of cytostatics is neutropenia, which remains a major risk factor for candidiasis and sepsis due to *Candida*—still the most important etiological factor in human fungal infections [3].

Azoles, of which the most popular representative is fluconazole, remain first-line drugs in most cases, both in local and disseminated candidiasis. They are fungistatic drugs and their mechanism of action is based on the inhibition of the enzyme involved in the synthesis of ergosterol—a component of the fungal cell membrane. Ergosterol deficiency results in changes in membrane permeability and inhibition of fungal cell growth, contributing to its death [4]. Due to its good safety profile, in addition to the ability to treat various forms of fungal infections, fluconazole is often used prophylactically, which is especially justified in patients with neutropenia, after organ transplantation, and in women with recurrent vulvovaginal candidiasis [5]. The above makes the emergence of fungal strains resistant to the drug in question more likely, contributing to numerous clinical problems. There are many mechanisms that can lead to the development of azole resistance. Among the most important are: point mutations in the gene encoding lanosterol 14α-demethylase (ERG11), ERG11 overexpression, reduced azole uptake, protein pumps ejecting xenobiotics from the cell, mutations in the ERG3 gene, and biofilm formation [6].

It has been observed that *Candida* spp. strains incubated with methotrexate (MTX) develop multi-drug cross-resistance to azoles [7,8,9,10]. Although the phenomenon was described in vitro, there is a lack of data on its occurrence in vivo among patients treated with cytostatics [10]. The mechanism of *Candida* cross-resistance to azoles is complex and results, among others, from the increase in the expression of the CaMDR1 gene (*Candida albicans* Multi Drug Resistance) under the influence of induction with methotrexate, but also with other not structurally or functionally related substances such as benomyl, diethyl malonate, or o-phenanthroline [7,8,9]. The described mechanism has been proven to be sufficient to induce resistance to some toxic substances in yeast [11]. It was also confirmed that CaMDR1 overexpression occurs in wild azole-resistant clinical strains [12]. Therefore, there is a potential risk of antimicrobial resistance in endogenous *Candida* spp. isolates in patients who have never been treated with antifungal drugs, but have been exposed to methotrexate. It is possible that other commonly used cytostatics may also reduce the sensitivity of clinical isolates of *Candida* spp. to some antifungal agents [10].

The danger of infection with resistant *Candida* strains in patients on cytostatic therapy is therefore related not only to the weakening of the body’s ability to fight infection under the influence of immunosuppressive treatment, but also to increased drug resistance of fungi, which entails difficulties in implementing proper therapy [1,2,13,14]. If the process of selection for multi-resistant strains actually takes place in people treated with cytostatics, then to combat diseases caused by such isolates, it is necessary to change the type of antimycotic to one that will ensure effective elimination of the pathogenic agent. The previous statement seems to be especially true in life-threatening situations, for example, due to developing sepsis, when the patient is treated empirically before the drug sensitivity of the microorganism is determined.

Due to chronic, long-term exposure to therapeutic doses of methotrexate, patients suffering from rheumatological diseases such as psoriatic arthritis or rheumatoid arthritis seem to be an appropriate group to look for the effect of cytostatics on the induction of drug resistance in endogenous *Candida* spp. in vivo. However, to minimize the risk of colonization with exogenous, multi-resistant *Candida* strains derived from hospital flora, neither the test nor the control groups should be exposed to such isolates. For this reason, patients undergoing outpatient therapy who have not been recently hospitalized and have never been exposed to azole drugs seem to be the appropriate group to examine research problem presented above.

The aim of the study was to assess whether patients taking methotrexate are more often than people not exposed to this substance colonized with *Candida* spp. strains presenting resistance to fluconazole. Such observation would indicate that patients treated with cytostatics develop antimicrobial cross- resistance among endogenous *Candida* isolates. We also estimated the frequency of oral colonization by *Candida* spp. in patients of Rheumatology Outpatient Clinic and Rheumatology Day Care Unit in comparison to cytostatic-naive population without rheumatological diseases, and searched for the presence of relationships between the prevalence of resistant *Candida* spp. strains with the obtained clinical data: sex, age, the weekly MTX dose, the cause and the duration of the MTX therapy.

## 2. Materials and Methods

The test group (52 individuals) consisted of patients from the Rheumatology Outpatient Clinic and the Rheumatology Day Care Unit of Wrocław Teaching Hospital, in outpatient treatment with methotrexate for at least 6 months, who have never been exposed to fluconazole. The control group (49 persons) consisted of individuals who have never been exposed to methotrexate or fluconazole, regardless of their medical status and comorbidities. Participants from both groups who were hospitalized for more than 24 h in the last 2 years were excluded from the study.

Oral swabs were taken from all subjects. The obtained material was cultured for 24 h at 37 °C on a liquid medium and Sabouraud Agar medium. Each strain was isolated and then identified using mass spectrometry—MALDI TOF. Minimal inhibitory concentrations (MICs) for fluconazole were determined for each strain using the microdilution method in RPMI 1640 liquid medium in accordance with the guidelines of the Clinical and Laboratory Standards Institute (CLSI) [15,16]. Suspensions of the tested strains (0.5–2.5 × 10^3^ CFU per mL) were applied to 96-well polystyrene plates with previously prepared serial dilutions of fluconazole in ranging concentration of 0.125 and 256 µg/mL. The plates were incubated for 24 h at 37 °C. To determine MICs, the optical density (OD) was read spectrophotometrically (BiochromAsys UVM 340) at a wavelength of 530 nm. All experiments were conducted in duplicate and included a strain growth control (positive control; K+) and a negative control (K−), which served as a medium sterility test. The MIC was considered as the concentration of fluconazole at which the growth inhibition of at least 50% of microorganisms (using the equation: (OD_well_-OD_K_^−^)/(OD_K+_-OD_K_^−^) × 100%) was detected. In the next step, we compared the MIC values in both groups and determined the percentage of fluconazole resistant isolates in each of them. The data were interpreted based on the clinical breakpoints recommended by the CLSI M60-Ed2 [16].

In addition, information on patients’ sex, age, the use of dentures, current methotrexate dose, duration and cause of methotrexate therapy was collected. The authors analyzed the obtained microbiological data comparing them with clinical information on individual patients. The most important data characterizing both studied groups are included in Table 1.

The study was conducted according to the guidelines of the Declaration of Helsinki and approved by the Bioethical Committee at the Wrocław Medical University (KB-191/20 approved on 4 April 2020).

## 3. Results

### 3.1. Identification

15.4% (8/52) of the test group samples and 30.6% (15/49) of the control group samples gave negative cultures. From the remaining ones, 1 to 4 strains of fungi were obtained (Table 2 and Table 3). Both in the study group and in the control group, *Candida albicans* was the most frequently isolated species. The composition of the identified fungal flora was less diverse in the test group. Moreover, control group samples significantly more often than samples from the group of patients exposed to MTX contained at least 2 species of fungi (32.4% (11/34) vs. 11.4% (5/44), respectively, RR = 2.85, 95% CI: 1.09–7.42, *p* = 0.03).

### 3.2. Minimum Inhibitory Concentration (MIC)

Both in the test group and in the control group, the majority of the grown strains were sensitive to fluconazole. The most common MIC value was x < 0.125 µg/mL. Four strains obtained from 4 patients of the test group (7.7% of samples) and 5 strains obtained from 5 patients of the control group (10.2% of samples) had MIC above 1 µg/mL. Figure 1 shows the MIC values determined for all isolated *Candida* strains.

According to the 2020 CLSI criteria, 5 strains derived from samples obtained from 4 patients of the test group (all belonging to the *C. krusei* species) and 2 strains from 2 patients of the control group (*C. krusei* and *C. albicans*) can be considered resistant to fluconazole [16]. Table 4 shows the characteristics of the samples from which the resistant strains described above were grown. Eight strains belonging to the *C. glabrata* species (4 in the control group and 4 in the test group) showed dose-dependent fluconazole sensitivity (SDD). The MIC value for a given strain was not significantly related to age, sex, cause and duration of MTX therapy or the presence of dentures. Detailed data on the characteristics of study participants, samples, and the obtained laboratory results are available in Appendix A.

All resistant strains in the study group occurred in samples obtained from patients taking MTX at a dose of at least 20 mg/week. In addition, the proportion of MIC results other than <0.125 µg/mL among all MIC results obtained for *Candida* strains from the study group was greater in the group of patients receiving MTX at a dose of at least 20 mg/week than in patients receiving MTX at lower doses (7/28 (25.00%) and 2/21 (09.52%) respectively), but this relationship did not reach statistical significance (RR = 2.63, 95% CI: 0.61–11.31, *p* = 0.20). More specific information on the MIC values for a given strain and the weekly dose of MTX are included in Table 5.

## 4. Discussion

*Candida* species are part of the natural fungal flora of the skin and mucosa, occurring even in 50–70% of the world’s population [3,13]. Nevertheless, under favorable conditions, especially in people with risk factors for fungal diseases, these microorganisms are responsible for symptomatic infections of various tissues and organs. Mucocutaneous candidiasis, which may manifest as oral candidiasis or vulvovaginal candidiasis, is the most common *Candida*-caused illness worldwide [17,18]. Moreover, *Candida* species are responsible for the majority of cases of fungemia and fungal sepsis [14]. Although *C. albicans* still remains the most commonly isolated species both in the case of mucocutaneous candidiasis and disseminated candidiasis, its role has been gradually declining in recent years. On the other hand, the percentage of infections caused by NCAC (Non-*Candida albicans Candida*), including species traditionally considered more resistant to standard antifungal drugs (e.g., *C. tropicalis*, *C. glabrata*, *C. kefyr*, *C.parapsilosis*, or *C. krusei*), is growing [3,13]. 

Factors contributing to symptomatic fungal infections include: neutropenia, neoplastic disease, diabetes, use of dentures, antibiotic therapy, corticosteroid therapy, administration of immunosuppressants, recent surgery, and hospitalization in the intensive care unit [2,19]. Some rheumatological diseases characterized by decreased saliva production (especially Sjögren’s syndrome; SjS) significantly increase the risk of oral candidiasis. In a 2011 study, oral candidiasis occurred in 87% of 30 qualified patients with SjS, while fluconazole-resistant strains accounted for 41% of all isolates [20]. The most frequently isolated species in this case was *C. albicans* (25/30 samples), but as many as 44% of the positive cultures had at least 2 *Candida* species—usually, *C. albicans* and *C. tropicalis*. *C. krusei* was present, always accompanied by 2 other species of the genus, in 2 out of 30 samples (6.7%).

Furthermore, certain specific conditions predisposing to oral candidiasis have been identified in SLE patients. These include: African-American descent, high disease activity, high white blood cells levels, history of recent bacterial infection, proteinuria, and the use of prednisone and immunosuppressants [21].

Previous studies also indicate a more frequent, compared to the healthy population, oral colonization by *Candida* species in patients suffering from rheumatoid arthritis, which is largely due to the impaired TH_17_-dependent immune response observed in this disease [22,23].

Taking into account the risk factors described above, the more frequent colonization of the oral cavity by fungi in the group of rheumatological patients, compared to the control group, which we observed in our study (84.6% and 69.5%, respectively, RR = 1.22; 95% CI: 0.98–1.52; *p* = 0.08) seems to be justified by the available literature [2,19,20,21,22,23,24].

As mentioned above, it seems that rheumatological diseases, as a result of decreased saliva production, specific immunological mechanisms and immunosuppressive action of drugs, promote oral fungal infections. In this study, it was observed that multispecies cultures occurred significantly more often among people who did not take MTX (control group) than among rheumatological patients for whom single-species cultures were more characteristic. Perhaps the higher incidence of oral fungal infections in rheumatological patients is at least in part related to disturbances in the oral microbiome, leading to a predominance of single *Candida* species, rather than two or more competing ones. Theoretically, such uncontrolled growth of one species could be responsible for the etiological mechanism of symptomatic oral candidiasis.

Cross-resistance, which has been the subject of numerous studies in recent decades, remains an important clinical problem in the treatment of diseases caused by pathogenic microorganisms. This term refers to a situation where one drug induces resistance to another drug or a group of drugs that have not exerted a selective pressure on the organism acquiring resistance [8,9,10,25]. A well-known example of cross-resistance occurring in bacteria is co-resistance to macrolides, lincosamides, and streptogramin B (MLSB), which may be caused by methylation of the large ribosome subunit encoded by the ermA, ermB, ermC or ermF genes [25].

For MTX and azoles, the best explained mechanism of cross-resistance is based on the methotrexate-induced increase in CaMDR1 gene expression found in *C. albicans*. This gene is responsible for the synthesis of the membrane pump (CaMdr1p) removing xenobiotics, including methotrexate and fluconazole, from yeast cells [26]. There are numerous studies describing this phenomenon [7,8,9,10,26]. The authors of a recent study, examining the ability of methotrexate to induce resistance to fluconazole, itraconazole, and voriconazole in *C. albicans* and *Meyerzyme guilliermondii*, observed a significant increase in the median MIC for fluconazole and voriconazole—in the case of the former species, as well as for fluconazole and itraconazole—in the case of the latter. The extent of cross-resistance to azoles under the influence of MTX induction shows significant species differences, which should be reflected in future studies of this problem. In the mentioned paper, after the induction process, all analyzed fungal strains showed resistance to fluconazole, but it should be noted that in the case of *M. guilliermondii*, 53% of the tested isolates were resistant even before the use of MTX [10].

It is not entirely certain whether this CaMDR1-dependent mechanism is present in other species of *Candida*. However, due to the fact that induction of azole resistance by methotrexate has been observed among NCAC species and the existence of genes homologous to CaMDR1 among other yeasts has been proven, such an assumption seems to be justified [9,10,26,27].

Despite relatively unambiguous laboratory data, there have been no studies assessing the risk of cross-resistance between MTX and azoles in vivo in patients treated with cytostatics [10]. According to the authors’ knowledge, this study is the first analysis of this type which investigates the effect of systemic methotrexate therapy on the occurrence of azole-resistant *Candida* strains in clinical conditions.

In our study, no significantly higher incidence of resistance to fluconazole among *Candida* strains from the group of patients treated with MTX, compared to the control group, was observed (7.7% vs. 4.1% of samples; RR = 1.88; 95% CI: 0.36–9.83; *p* = 0.45). All resistant isolates detected in oral swabs of MTX-treated patients and 1 of 2 specimens containing resistant strains obtained from control group patients were *C. krusei*. It is a species showing natural resistance to fluconazole, which is predominantly caused by reduced sensitivity of lanosterol 14α-demethylase to the discussed drug [28]. Perhaps the greater prevalence of *C. krusei* in the study group than in the control group (7.7% vs. 2.0% of samples; RR = 3.77; 95% CI: 0.44–32.56; *p* = 0.23) results from more frequent colonization with isolates of this species among rheumatological patients compared to the rest of the population, or from the lower sensitivity of *C. krusei* to the fungistatic effects of methotrexate. However, these hypotheses are not sufficiently supported by the available literature.

Methotrexate is a cytostatic drug belonging to the group of antimetabolites. Its mechanism of action is based on the inhibition of tetrahydrofolate dehydrogenase, which results in impaired metabolism of folic acid. It shows a weak antifungal activity and also, when used in combination with classic antimicotics, it exerts a synergistic antifungal effect [29]. This drug is widely used in numerous branches of medicine. Some of the indications for systemic methotrexate therapy include: solid tumors (lung cancer, breast cancer, ovarian cancer, testicular cancer, head and neck cancers, bone sarcomas), acute leukemias, lymphomas, psoriasis and psoriatic arthritis, rheumatoid arthritis and systemic lupus erythematosus [1]. Due to the fact that methotrexate is a myelotoxic drug, its use may result in neutropenia, thus further increasing the risk of disseminated fungal infection [2]. Methotrexate resistance may be caused by increased ejection of xenobiotic from the neoplastic cell by overexpression of the membrane pump, which has been observed in some neoplasms [30].

The proper dose of MTX is determined individually, depending on the indications, general condition, and blood counts. In psoriasis and rheumatological diseases such as RA, the optimal dose is usually between 10–25 mg/week. The doses used in the treatment of solid tumors and leukemias are much higher than those administered in autoimmune diseases. A single antineoplastic dose can be low (100 mg/m^2^), medium (up to 500–1000 mg/m^2^), or high (>500–1000 mg/m^2^). In this case, chemotherapy is usually given as several cycles over a period of few months [31,32]. In our study, the lowest MTX dose was 7.5 mg/week and the highest was 25 mg/week. The most frequently taken MTX dose, which was also the median dose, was 20 mg/week. We did not find any significant relationship between the MIC value and the dose or duration of MTX therapy; however, it is worth mentioning that in the study group, all resistant strains and all strains with MIC > 0.25 µg/mL occurred in patients with a weekly dose of MTX higher or equal to 20 mg. This could potentially indicate that systemic MTX therapy exerts a selective pressure on endogenous *Candida* strains, and it is the greater the higher the MTX dose. Due to the fact that patients taking methotrexate for the treatment of rheumatological diseases are chronically exposed to relatively low doses of this drug compared to hematology and oncology patients, the effect of inducing azole resistance in this group may be proportionally lower [1,33]. To test this hypothesis, future studies should be conducted on a larger group of patients, in a wider range of MTX doses, including as high as those used in oncological and hematological diseases. If this phenomenon indeed occurs in clinical conditions, the selection of an appropriate antifungal therapy will contribute to a more effective treatment of yeast infections, potentially improving the clinical prognosis.

## 5. Conclusions

*Candida* species occurred more frequently in oral swabs from patients with rheumatological diseases treated with methotrexate than in the control group. In both groups of patients, the majority of the cultured strains showed sensitivity to fluconazole. All resistant strains in the study group occurred in samples obtained from patients taking MTX at a dose of at least 20 mg/week. Despite slightly higher incidence of fluconazole-resistant strains among patients on systemic methotrexate therapy, we found no solid evidence to support the idea that methotrexate induces resistance to azoles among endogenous *Candida* strains in patients with rheumatological diseases. No significant associations were observed between the MIC value for a given strain, age, sex, cause or duration of MTX treatment, and the presence of dentures. Due to the existence of strong theoretical premises and possible high clinical benefits resulting from better adjustment of antifungal therapy to the anticipated sensitivity of the microorganism to antimycotics, in the opinion of the authors, it is necessary to conduct a similar study on the population of patients receiving methotrexate in doses higher than those used in the treatment of rheumatological diseases.

## Figures and Tables

**Figure 1 antibiotics-10-01302-f001:**
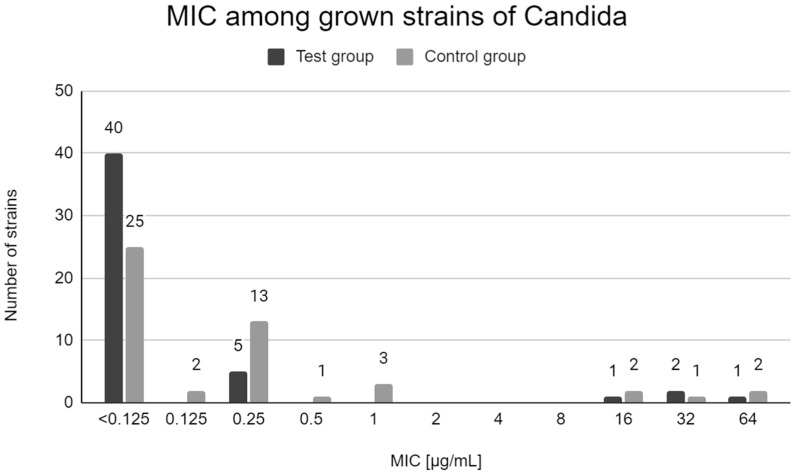
Distribution of MICs of fluconazole among *Candida* strains in test group patients and control group patients.

**Table 1 antibiotics-10-01302-t001:** Characteristics of the study participants. RA—Rheumatoid arthritis, PsA—Psoriatic arthritis, SLE—systemic lupus erythematosus, AS—ankylosing spondylitis, SpA—spondyloarthritis, JIA—juvenile idiopathic arthritis, SSc—systemic sclerosis, GPA—granulomatosis with polyangiitis, SjS—Sjögren’s syndrome.

KERRYPNX	Test Group (*n* = 52)	Control Group (*n* = 49)
Age (years), mean (SD)	52.21 (13.21)	51.45 (14.97)
Gender	Male: 18/52 (34.6%)	Male: 25/49 (51.0%)
	Female: 34/52 (65.4%)	Female: 24/49 (49.0%)
Dentures presence	13/52 (25.0%)	9/49 (18.4%)
Total duration of methotrexate therapy (years), mean (SD)	5.90 (4.47)	-
Methotrexate dose (mg/week), mean (SD)	19.09 mg, 20 mg (5.24 mg)	-
Cause of methotrexate therapy	RA: 34	-
	PsA: 7	
	RA/PsA: 1	
	SLE: 3	
	AS: 2	
	SpA: 1	
	JIA: 1	
	SSc: 1	
	GPA: 1	
	SjS: 1	

**Table 2 antibiotics-10-01302-t002:** Cultured species of fungi in samples taken from the test group.

Single-Species Samples		Multiple-Species Samples	
Species	Number of Samples	Species	Number of Samples
*C. albicans*	33	*C. albicans + C. dubliniensis*	0
*C. dubliniensis*	0	*C. albicans + C. glabrata*	2
*C. glabrata*	1	*C. albicans + C. inconspicua*	1
*C. kefyr*	1	*C. albicans + C. krusei*	1
*C. krusei*	2	*C. albicans + C. tropicalis*	0
*C. lusitaniae*	1	*C. albicans + Rhodotorulla mucilaginosa*	0
*C. tropicalis*	0	*C. krusei + C. glabrata*	1
*Hanseniaspora uvarum*	0	*C. glabrata + C. tropicalis*	0
*Wickerhamomyces anomalus*	1	*C. albicans + C. farmata + C. glabrata*	0
		*C. albicans + C. parapsilosis + C. zeylanoides*	0
		*C. tropicalis + C. Glabrata + Yarnodria lipolytica*	0
		*C. albicans + C. dubliniensis + C. krusei + C. tropicalis*	0

**Table 3 antibiotics-10-01302-t003:** Cultured species of fungi in samples taken from the control group.

Single-Species Samples		Multiple-Species Samples	
Species	Number of Samples	Species	Number of Samples
*C. albicans*	16	*C. albicans + C. dubliniensis*	2
*C. dubliniensis*	4	*C. albicans + C. glabrata*	1
*C. glabrata*	0	*C. albicans + C. inconspicua*	0
*C. kefyr*	0	*C. albicans + C. krusei*	0
*C. krusei*	0	*C. albicans + C. tropicalis*	2
*C. lusitaniae*	1	*C. albicans + Rhodotorulla mucilaginosa*	1
*C. tropicalis*	1	*C. krusei + C. glabrata*	0
*Hanseniaspora uvarum*	1	*C. glabrata + C. tropicalis*	1
*Wickerhamomyces anomalus*	0	*C. albicans + C. farmata + C. glabrata*	1
		*C. albicans + C. parapsilosis + C. zeylanoides*	1
		*C. tropicalis + C. Glabrata + Yarnodria lipolytica*	1
		*C. albicans + C. dubliniensis + C. krusei + C. tropicalis*	1

**Table 4 antibiotics-10-01302-t004:** Characteristics of samples containing *Candida* strains showing resistance (R) to fluconazole. S—fluconazole-sensitive strain, SDD—dose-dependent sensitivity, RA—Rheumatoid arthritis, SpA—spondyloarthritis.

Sample	Species	MIC (µg/mL) (Susceptibility Status)	Gender (Age), Duration of MTX Therapy, Weekly Dose, Cause of Therapy, Dentures Presence
T11, Test group	*C. glabrata* *C. krusei* ^1^	32 (SDD)	Male (77), 10, 20 mg, RA, dentures present
		<0.125 (R)	
T12, Test group	*C. albicans* *C. krusei* ^1^	<0.125 (S)	Male (66), 15, 25 mg, RA, dentures present
		64 (R)	
T41, Test group	*C. krusei* ^1^	16 (R)	Female (66), 2, 25 mg, RA, dentures present
T49, Test group	*C. krusei* ^1^ *C. krusei* ^1^	<0.125 (R)	Female (45), 6, 20 mg, SpA, dentures absent
		<0.125 (R)	
C40, Control group	*C. albicans* *C. dubliniensis* *C. krusei* ^1^ *C. tropicalis*	0.25 (S)	Male (64), dentures present
		0.25 (S)	
		64 (R)	
		0.25 (S)	
C43, Control group	*C. albicans*	64 (R)	Male (73), dentures present

^1^ *C. krusei* is intrinsically resistant to fluconazole and therefore the MIC values do not reflect the actual sensitivity status [16].

**Table 5 antibiotics-10-01302-t005:** Relationship between the MIC value for fluconazole among the cultured *Candida* strains of the studied group and the weekly dose of methotrexate.

MTX Dose (mg/Week)	Number of *Candida* Strains with Specific MIC Value (µg/mL)/All *Candida* Strains in the Dose Category					Number of Patients
	64	32	16	0.25	<0.125	
25	1/16	1/16	1/16	2/16	11/16	18
20	0/12	1/12	0/12	1/12	10/12	12
17.5	0/1	0/1	0/1	1/1	0/1	1
15	0/13	0/13	0/13	0/13	13/13	15
12.5	0/1	0/1	0/1	0/1	1/1	1
10	0/5	0/5	0/5	1/5	4/5	4
7.5	0/1	0/1	0/1	0/1	1/1	1

## Data Availability

All data generated or analyzed during this study are included in this published article and Appendix A.

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
