# Peer review of "Does Systemic Methotrexate Therapy Induce Azole Resistance among Endogenous Candida Strains?"

_antibiotics, 2021, doi:10.3390/antibiotics10111302_

Round 1

Reviewer 1 Report

In this study, the authors examined the cross-resistance to methotrexate and azoles in Candida spp. strains isolated from patients treated with cytostatics. The clinical study seems to be well designed and conducted, and manuscript is well written. This reviewer thinks the manuscript is acceptable for publication in the journal without revision.

Author Response

Point 1. In this study, the authors examined the cross-resistance to methotrexate and azoles in Candida spp. strains isolated from patients treated with cytostatics. The clinical study seems to be well designed and conducted, and manuscript is well written. This reviewer thinks the manuscript is acceptable for publication in the journal without revision.

Response 1. Thank you very much for your review. I encourage you to read the revised version of the manuscript.

Reviewer 2 Report

Section, Introduction

Please provide more specific information about azoles

Possibly the paragraph (lines 187-201) could be moved to this section to complete the necessary information

Line 69-72

Please rephrase or split into two sentences for better meaning

Line 176

Please provide the meaning of WBC

Section, Discussion

This section has plenty of literature information, please consider moving some paragraphs to provide more discussion on data and results

Lines 273-277

(…. We found no solid evidence….)

Please consider rearranging the title of the manuscript (it is misleading)

Author Response

We greatly appreciate your review. It turned out to be extremely helpful and definitely enriched our work.

Point 1. Section, Introduction; Please provide more specific information about azoles. Possibly the paragraph (lines 187-201) could be moved to this section to complete the necessary information

Response 1. The authors followed the reviewer's suggestions by moving the paragraph on the characteristics of azoles from the discussion to the introduction.

Point 2. Line 69-72;Please rephrase or split into two sentences for better meaning.

Response 2. The authors followed the reviewer's suggestions by splitting the sentence in question into two sentences.

Point 3. Line 176; Please provide the meaning of WBC

Response 3. The abbreviation WBC (white blood cells) has been replaced with the full name

Point 4. Section, Discussion; This section has plenty of literature information, please consider moving some paragraphs to provide more discussion on data and results. 

Response 4. The authors followed the reviewer's suggestions by moving the paragraph on the characteristics of azoles from the discussion to the introduction and by adding paragraphs on methotrexate doses and single-species samples .

Point 5.Lines 273-277;(…. We found no solid evidence….);Please consider rearranging the title of the manuscript (it is misleading).

Response 5.  The authors followed the reviewer's suggestions by changing the title of the manuscript to: “Does systemic methotrexate therapy induce azole resistance among endogenous Candida strains?”

Reviewer 3 Report

It has been reported in different previous papers that Candida spp. strains incubated with methotrexate (MTX) develop multi-drug cross-resistance to azoles due to higher expression of the CaMDR1 gene. These papers were based on laboratory data, but there is no evidence yet assessing the risk of cross-resistance between MTX and azoles in patients treated with fluconazole and other cytostatics.

The aim of the study was to assess whether patients treated with cytostatics develop antimicrobial cross-resistance among endogenous Candida strains indeed. The study was focused especially whether patients taking MTX are more often colonized with Candida spp. strains presenting resistance to fluconazole than people not exposed to this substance

The conclusion is that there is no solid evidence to support the idea that methotrexate induces resistance to azoles among endogenous Candida strains in patients with rheumatological diseases. To reconcile previous and current data, authors propose that the MTX doses could be crucial. They assume that patients taking MTX for treatment of rheumatological diseases are exposed to relatively low doses of the drug in comparison hematology and oncology patients.

In that way, the effect of inducing azole resistance in this group may be proportional, and according to that the possible induction of azole resistance by MTX should be further studied including patients treated with higher doses of MTX.

The main concern to this proposal is that the doses of the patients studied in the present work is not given. The paper can be accepted if those data are included and discussed in comparison to actual references of doses in hematology and oncology patients. These data are crucial, much more crucial than data included in the supplementary data. Otherwise, the manuscript cannot be accepted.

Other minor points to be addressed provided than doses of patients are included

Format of the References should be normalized and unified. Some of them are not completed, and all of them should be easily found (complete volume, pages and year or alternatively Doi).

Line 120: control group samples contained at least 2 species of fungi significantly more often than samples from the group of patients exposed to MTX. Any hypothesis or correlation with the MYX treatment?

Author Response

Dear Sir/ MadamDear Sir / Madam
We greatly appreciate your review. It turned out to be extremely helpful and definitely enriched our work.

Point 1. (…)The main concern to this proposal is that the doses of the patients studied in the present work is not given. The paper can be accepted if those data are included and discussed in comparison to actual references of doses in hematology and oncology patients. These data are crucial, much more crucial than data included in the supplementary data. Otherwise, the manuscript cannot be accepted.

Response 1. Following your suggestion, we have completed the data on the MTX doses. At the beginning, however, I would like to mention that originally our study was not designed to accurately investigate the relationship between the methotrexate dose or the concentration in the patient's tissues with the presence of azole-resistant strains, but rather it was intended to indicate whether the postulated phenomenon is clinically observable in rheumatological patients. The authors do not have data allowing to track cumulative exposure to methotrexate, and the only information we have obtained on this subject is the duration of treatment with methotrexate and the patient's dose of methotrexate at the time of qualifying for the study (added to the supplement). Nevertheless, the inclusion of data on the MTX doses allowed us to describe new threads regarding the influence of MTX on the MIC values (lines 168-177, Table 1, Table 5). In addition, the discussion now includes more detailed information on the dosing of MTX in haematological and oncological diseases and compares them with the MTX doses taken by the study participants (please see the new paragraphs in the discussion).

Point 2. Line 120: control group samples contained at least 2 species of fungi significantly more often than samples from the group of patients exposed to MTX. Any hypothesis or correlation with the MYX treatment?

Response 2. As mentioned in the discussion of this paper, it seems that rheumatological diseases as a result of decreased saliva production, through immunological mechanisms and immunosuppressive action of drugs favor fungal infections. In this study, it was observed that multispecies cultures occurred significantly more often among people who did not take MTX (control group) than among rheumatological patients for whom single-species cultures were more characteristic. Perhaps the higher incidence of oral fungal infections in rheumatological patients is at least in part related to disturbances in the oral microbiome, leading to a predominance of single Candida species, rather than two or more competing ones. Theoretically, such uncontrolled growth of one species could manifest itself as symptomatic oral candidiasis. We have now included the above information in the discussion .

Point 3. Format of the References should be normalized and unified. Some of them are not completed, and all of them should be easily found (complete volume, pages and year or alternatively Doi).

Response 3. The authors followed the reviewer's suggestions by unifying the literature as required.

Round 2

Reviewer 2 Report

revision accepted

Reviewer 3 Report

I think the new title is appropriate. According the reply letter, I realize that the study was not originally designed to accurately investigate the relationship between the MTX dose and the presence of azole-resistant strains, but the data re-conducted the discussion. Hence, I understand that authors do not have the complete set of data allowing to track cumulative exposure to MTX, but the partial information added to the modified version added to the supplement is somehow interesting to support the hypothesis of the dependence with the MTX dose. The question at the title is convenient, as the hypothesis is not proven yet, and it should be further investigated.

I think that the inclusion of data on the MTX doses at tables 1, 4 and 5, as well as the lines 168-177 has improved and focused the paper. Other replies at the reply letter and the subsequent modifications throughout the manuscript are also pertinent.